

# Construction of applied talents training system based on machine learning under the background of new liberal arts

Fei Tang

School of Architecture and Art Design, University of Science and Technology Liaoning, Anshan, Liaoning, China

## ABSTRACT

The development of the new liberal arts field places emphasis on the integration of disciplines such as humanities, engineering, medicine, and agriculture. It specifically highlights the incorporation of new technologies into the education and training of liberal arts majors like economics, law, literature, history, and philosophy. However, when dealing with complex talent data, shallow machine learning algorithms may not provide sufficiently accurate evaluations of the relationship between input and output. To address this challenge, this article introduces a comprehensive evaluation model for applied talents based on an improved Deep Belief Network (DBN). In this model, the GAAHS algorithm iteratively generates optimal values that are utilized as connection weights and biases for the restricted Boltzmann machines (RBM) in the pre-training stage of the DBN. This approach ensures that the weights and biases have favorable initial values. Moreover, the paper constructs a quality evaluation index system for creative talents, which consists of four components: knowledge level, innovation practice ability, adaptability to the environment, and psychological quality. The training results demonstrate that the optimized DBN exhibits improved convergence speed and precision, thereby achieving higher accuracy in the classification of applied talent evaluations.

Corresponding author
Fei Tang, tangfei_lnkjdx@163.com

## INTRODUCTION

Artificial intelligence technology has many utilizations in the field of personnel training, especially mining information from data itself, depending on the data processing ability of artificial intelligence technology. An artificial neural network (ANN) has a self-learning ability. It does not need to determine the weight of each index, which avoids the problem that the index weight is greatly affected by subjective factors. It can achieve a highly nonlinear mapping from input to output, so it is widely used. Traditional clustering algorithms can be divided into segmentation clustering, density clustering, and propagation based methods. Under big data background, using massive data resources and highlighting quantitative analysis is a significant development direction of innovative talent training research (*Lai & Yu, 2021*; *Jing et al., 2022*). Therefore, it is important to enhance the effect of talent classification by collecting the data of scientific and technological innovation talents and then mining and analyzing the sample data. The

existing research shows that the clustering algorithm is an effective method for studying classification problems in data mining. Traditional clustering algorithms can be divided into segmentation clustering, density clustering, and propagation-based methods (*Xu & Tian, 2015*). A density peak clustering algorithm (DPC) proposed by *Rodriguez & Laio (2014)* has the characteristics of fast computing speed and no iteration, which can describe the data distribution well. It has lower complexity than the general k-means algorithm. Although the DPC algorithm has apparent advantages, it still has some limitations in processing high-dimensional data and classifying non-central points. Aiming at the shortcomings of the DPC algorithm, many scholars have improved the DPC algorithm in recent years. *Rehman & Belhaouari (2022)* combined the advantages of the DPC algorithm and the Chame-Leon algorithm to propose the E_CFSFDP algorithm. Although it avoided clustering a class containing multiple density peaks into multiple classes, it required much computation and was not conducive to processing high-dimensional data. *Jiang et al. (2020)* proposed two fast-density peak algorithms, KNN-DPC based on the KNN sample allocation strategy, which has good robustness to noise data, but because the clustering process of the algorithm is the same as that of DPC, the defects of the DPC algorithm still exist in the algorithm.

There are abundant research results on the teaching of big data courses. Still, most of them regard big data as the background of The Times or technical means and discuss the role of big data in teaching management, teaching evaluation, teaching philosophy and teaching mode. There are opportunities and challenges brought by teaching techniques, *etc*., while teaching and research achievements in big data analysis courses are relatively few (*Wang, Hao & Lu, 2023*). The essence of the new liberal arts is to reorganize the traditional liberal arts, break professional barriers and disciplinary boundaries, integrate emerging technologies such as big data and artificial intelligence into professional learning and training, and change the talent training mode (*Li & Xu, 2023*). Because data collection, mining, analysis and presentation require specific programming abilities, this course has slightly different requirements for teachers and students than other courses. Especially for humanities and social sciences, students face some unique challenges in the teaching process. Compared with conventional thinking, creative thinking is an advanced thinking activity with originality, flexibility and risk. It is necessary to continuously input knowledge in different fields and actively carry out individual intelligent development and scientific thinking training to improve the quality of thinking (*Mernagh & Jennings, 2019*). Only in this way can we break through the inherent thinking and obtain the results of creative thinking in repeated thinking. The internal factors restricting creative thinking activities are complex and diverse. Each constituent element is a multi-level subsystem with different characteristics and functions in the whole process of creative activities (*Guan & Zheng, 2021*). Among them, the main basic elements include knowledge, concept, problem consciousness, thinking quality, *etc*. Overall, the essence of the evaluation of College Students' comprehensive quality is to evaluate the various performances of students during the whole university period. The assessment of colleges and universities is often based on different educational policies and social needs. Therefore, the evaluation results of

students' comprehensive quality can be used to test the quality of applied talents training in colleges and universities. This can help to select applied talents.

The main contributions of this article are as follows:

(1) Design a comprehensive evaluation model for creative talents based on improved DBN. In this model, the optimal result value generated by the iteration of the GAAHS algorithm is used as the connection weight and bias of RBM so that the weight and bias in the pre-training stage of DBN have good initial values.

(2) Construct the comprehensive quality evaluation index of practical talents under the background of new liberal arts, and realize the high-precision classification of practical talents through model training.

## LITERATURE REVIEW

The theoretical research on cultivating applied talents mainly concentrates on the training stage. *Xue & Li (2022)* put forward opinions on the problems existing in applied talent training in China from the perspective of industry-university research integration. However, there is relatively little research on the classification of applied talents. *Xu (2021)* explored the training classification mode of practical talents in colleges and universities in the flipped classroom under the background of new liberal arts. *Jia et al. (2020)* proposed a new method of policy classification to improve the implementation effect of policies in classifying innovative talents. Most of the above-related research is based on the theory of the qualitative research stage. The relevant quantitative research is very few, which leads to the lack of quantification and refinement of the research on the classification of applied talents, and cannot fully excavate the data information of applied talents to guide the classification of applied talents in practice.

To effectively identify and utilize talents, it is paramount to conduct scientific evaluations and assessments. Thus, establishing a robust talent evaluation mechanism and employing efficient evaluation methods throughout the entire talent lifecycle, including talent acquisition, training, selection, and utilization, holds immense significance. *Gordon & Rajagopalan (2016)* proposed a talent quality evaluation system design method based on evidence reasoning. Based on the basic theory of evidence reasoning method of hybrid programming technology, this method designs three modules, namely evaluation analysis, data management and index system management, to form the framework of the talent evaluation system and realize talent quality evaluation through quantitative and qualitative indicators. However, the error of the evaluation results obtained by this method is large, and the evaluation accuracy is low; *Wu, Hao & Kim (2017)* tried to explore the production of "The Belt and Road Initiative" driven animation projects by crowdsourcing mode, where the Delphi method is mainly used for investigation. But the system evaluation designed by this method takes a long time and has the problem of low evaluation efficiency; *Al Aina & Atan (2020)* put forward a design method for a talent quality evaluation system based on distance measurement. This method selects a talent quality evaluation index, establishes a talent quality evaluation model, sorts and analyzes the schemes obtained by the model through the distance measurement method and realizes the design of a talent quality comprehensive evaluation system, which takes a long time to

sort the schemes, thus reducing the evaluation efficiency of the system. Most of the above related research is based on the theory of qualitative research stage, the relevant quantitative research is very few, which leads to the lack of quantification and refinement of the research on the classification of talents, and cannot fully excavate the data information of them, so as to guide the classification of creative talents in practice.

With the rapid development and broad application of AI technology, more and more artificial intelligence technology has been introduced and adopted in the field of talent evaluation. The evaluation model based on artificial intelligence technology has been continuously developed and utilized. *Ye (2016)* constructed the evaluation index system of the scientific research ability of colleges and universities from two aspects of research input and output. Combined with the advantages of fuzzy theory and neural network, the proposed TS model scientific research ability evaluation is based on fuzzy neural network. The analysis results show that the model can accurately predict the status of the scientific research ability of different universities. *Chang & Li (2018)* constructed an evaluation model of innovative talents in a cultural industry based on BP neural network and conducted case analysis and verification. Compared with traditional evaluation methods, this model effectively avoided the influence of subjective factors in the evaluation process. *Yan & Zhao (2018)* proposed an evaluation model based on a fuzzy neural network, which more truly simulates the human brain's processing process of external information. However, with the deepening of research and the continuous growth of talent data, more and more factors are affecting the evaluation. At this stage, most of the evaluation methods based on artificial intelligence are shallow learning algorithms, which cannot process complex data.

## APPLIED TALENTS COMPREHENSIVE EVALUATION SYSTEM BASED ON OPTIMIZED DBN

### Overall design

The system has the capability to seamlessly integrate with existing student management systems in colleges and universities, enabling data sharing between the two. By retrieving relevant indicator information from the student management system, the system eliminates the need for duplicate data entry, minimizes redundancy, and prevents data waste. This integration not only reduces the workload but also enhances efficiency and accuracy simultaneously. Additionally, the system aids managers and decision-makers in effectively educating and managing students, while providing reliable references for adjusting talent training programs. The overall module design is shown in Fig. 1.

As a counsellor, the system has six sub-modules: evaluation index management, index weight setting, evaluation data management, comprehensive quality calculation, statistical query, and cluster analysis. Among them, the statistical query module to carry out various statistics on the comprehensive quality evaluation of college students, queries and displays according to different screening conditions, can count and query all the data information of all students, and can export and print operations; in addition, the cluster analysis module is based on the parallelized GAAHS-DBN algorithm to mine the evaluation

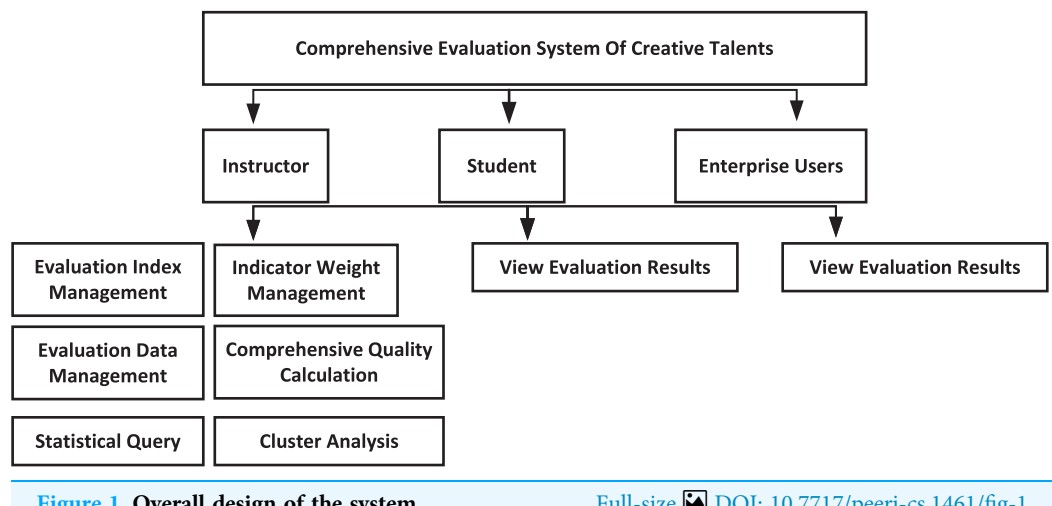

**Figure 1 Overall design of the system.**

results, and obtain the information beneficial to student's personal development and school education management.

As students access the system with their assigned roles, they gain access to their evaluation data, which encompasses index data, comprehensive evaluation data, significance and grade rankings, and quality levels. This provision enables students to assess their current standing and serves as motivation to improve their overall quality. Similarly, employers and recruitment agencies, accessing the system as enterprise users, can review the evaluation results of students. This feature facilitates a convenient means for users to grasp the collective quality of students and establish specific screening criteria based on the talent requirements. Consequently, they can effectively select the desired candidates according to their specific needs.

## Optimization of the DBN model

### DBN model

The optimized DBN model is a combinatorial optimization method which combines the global adaptive tuning Harmony Search (GAAHS) algorithm with the Deep Belief Network (DBN) model. With the help of the worldwide optimization ability of the GAAHS algorithm, feature extraction is carried out for each RBM unsupervised training in the DBN model. The connection weights and node offsets between the hidden layer and the explicit layer are optimized in the corresponding range. The optimal result value generated by the iteration of the GAAHS algorithm is taken as the connection weight and bias of RBM. Thus, the weights and biases of DBN have good initial values in the pre-training stage.

DBN is a generation model composed of multiple RBMs, each RBM contains a set of input units, and the deeper features are learned by a greedy layer-by-layer training method (*Ling et al., 2015*; *Hong et al., 2022*). Based on the model structure, this article introduces the GAAHS algorithm into DBN and proposes a DBN based on the GAAHS algorithm (GAAHS-DBN). The initial weights of RBM are obtained by GAAHS search. Then, the hidden layer is trained by using the CD algorithm to get the hidden layer and reconstruct

the explicit layer, and the network is trained layer by layer. Finally, a deep-seated DBN model is obtained.

In the basic harmony search (HS) algorithm, there are two main ways to generate new harmony (*Luu, Phien & Anh, 2021*; *Zhang et al., 2019*): (1) generate randomly in the solution space; (2) Select harmony from memory. Both of them adjust the step size BW to adjust and optimize the tone, so the reasonable step size assumes a significant part in the search mechanism of the HS algorithm.

When a small bandwidth (BW) value is utilized, the local search capability of the algorithm is strengthened. However, this can result in the generation of new harmonies being concentrated in a limited area, making it easier for the algorithm to become trapped in a local optimum. Additionally, at the initial stages of the search, the harmony memory may lack sufficient diversity in its solution vectors. On the other hand, using a large BW value helps to prevent the algorithm from getting stuck in local optima to some extent. However, this comes at the cost of reducing the algorithm's local search ability during the later stages of the search process.

Hence, the value of BW plays a crucial role in influencing both the global and local search abilities of the harmony search algorithm.

### Modeling steps

The modelling process of the GAAHS-DBN model is shown in Fig. 2, and its steps are elaborated below.

Step 1: initialize the structure and parameters of RBM in DBN: including the number of hidden layer and explicit layer nodes, learning rate and iteration times. After the network structure is determined, the harmony is coded, and the dimension of each harmony is calculated.

Step 2: initialize parameters of the GAAHS algorithm. Within the search range of harmony individuals, HMS harmony is generated randomly in a cycle, and the fitness value of each harmony is calculated to complete the initialization of the harmony memory database.

Step 3: optimize the initial weight and bias of RBM.

After generating a set of new weights and offsets, the training data is used for training. After training, the reconstruction error is calculated according to the reconstructed value and actual value of RBM. That is, the fitness is calculated; If the new fitness is better than the worst fitness, the new harmony will replace the one with the worst. Otherwise, it will remain unchanged; If the termination condition is satisfied, the algorithm will be terminated to complete the optimization. Otherwise, it will enter the next iteration process.

Step 4: The optimal harmony is used as the initial weight and bias of RBM, and the Contrastive Divergence algorithm (CD) is used to conduct unsupervised pre-training of RBM (*Krause, Fischer & Igel, 2018*; *Romero et al., 2019*).

Suppose that the number of neurons in the visual layer and a hidden layer of RBM is m and n. The state vector of visual layer v is $v = (v_1, v_2, v_{i,...,}, v_m)$, where $v_i$ represents the value of the ith neuron, and the bias vector is $a = (a_1, a_2, a_i, ..., a_m)$, where $a_i$ represents the bias of the i-th neuron; The state vector of hidden layer h is $h = (h_1, h_2, h_j, ..., h_n)$, $h_j$

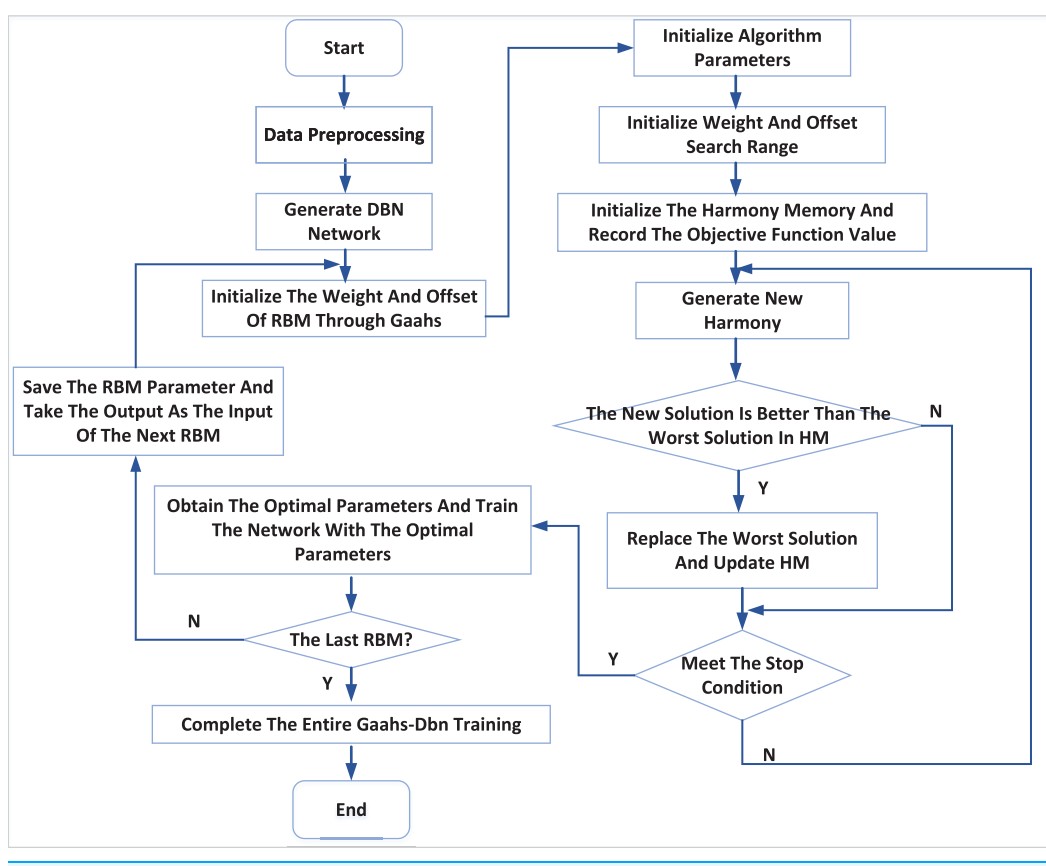

**Figure 2 DBN optimization process.**

represents the value of the j-th neuron. The bias vector is $b = (b_1, b_2, b_j, \ldots, b_n)$, where $b_j$ represents the bias of the j-th neuron, $w = (w_{ij})$ represents the connection weight between the i-th neuron in the visible layer and the j-th neuron in the hidden layer. Then for sample $(v, h)$, the gradient term of $\theta = \{w_{ij}, a_i, b_j\}$ is as follows:

$$\frac{\partial \log p(v)}{\partial w_{ij}} = \langle v_i h_j \rangle_{\text{data}} - \langle v_i h_j \rangle_{\text{model}} = P(h_j = 1|v)v_i - \sum_v P(v)P(h_j = 1|v)v_i \tag{1}$$

$$\frac{\partial \log p(v)}{\partial a_i} = v_i - \langle v_i \rangle_{\text{model}} = v_i - \sum_v P(v)v_i \tag{2}$$

$$\frac{\partial \log p(v)}{\partial b_j} = \langle h_j \rangle_{\text{data}} \langle h_j \rangle_{\text{model}} = P(h_j = 1|v) - \sum_v P(v)P(h_j = 1|v) \tag{3}$$

Among them, $\langle \cdot \rangle_{\text{data}}$ represents the expectation of the training sample, and $\langle \cdot \rangle_{\text{model}}$ represents the expectation of the model itself. $\langle \cdot \rangle_{\text{model}}$ is complex in the actual solving process, which seriously affects the algorithm's efficiency, so k-step Gibbs sampling is used to complete the parameter update (*Wang et al., 2020*). The k-step Gibbs sampling process is as follows:

(1) For the sample $(v, h) \in D$, the 0-th sampling is set as $v^{(0)} = v$.

(2) On the sample of the v and h alternating sampling: $h0 \sim P(h|v_0), v_1 \sim P(v|h_0), h_1 \sim P(h|v_1,.)v_2 \sim P(v|h_1), \ldots, h_k \sim P(h|v_k), \ v_{k+1} \sim P(v|h_k)$.

Using the result of k-step Gibbs sampling to approximate the parameter $\theta = \{w_{ij}, a_i, b_j\}$ gradient term:

$$\frac{\partial \log p(v)}{\partial w_{ij}} \approx P\left(h_j = 1|v^{(0)}\right)v_i^{(0)} - P\left(h_j = 1|v^{(0)}\right)v_i^{(k)} \tag{4}$$

$$\frac{\partial \log p(v)}{\partial a_i} \approx v_i^{(0)} - v_i^{(k)} \tag{5}$$

$$\frac{\partial \log p(v)}{\partial b_j} \approx P\left(h_j = 1|v^{(0)}\right) - P\left(h_j = 1|v^{(k)}\right) \tag{6}$$

In a given set of states $(v, h)$, the CD algorithm is used to learn the parameter update criteria of RBM at each layer as follows:

$$w'_{ij} = w_{ij} + \frac{\partial \log p(v)}{\partial w_{ij}} = w_{ij} + \varepsilon\left(P\left(h_j = 1|v^{(0)}\right)v_i^{(0)} - P\left(h_j = 1|v^{(0)}\right)v_i^{(k)}\right) \tag{7}$$

$$a'_i = a_i + \frac{\partial \log p(v)}{\partial a_i} = a_i + \varepsilon\left(v_i^{(0)} - v_i^{(k)}\right) \tag{8}$$

$$b'_j = b_j + \frac{\partial \log p(v)}{\partial b_j} = b_j + \varepsilon\left(P\left(h_j = 1|v^{(0)}\right) - P\left(h_j = 1|v^{(k)}\right)\right) \tag{9}$$

Among them, $\varepsilon$ represents the learning rate.

The solving process of parameters $w_{ij}, a_i, b_j$ is repeated until the final $w_{ij}{}', a'_i, b'_j$ are obtained after N iterations of the update, and the output value of the hidden layer is calculated, which is used as the input data of the visible layer of RBM of the next layer.

Step 5: Repeat Step 3 to Step 5 until all RBMs are trained, and then fine-tune the network parameters.

# MODEL TRAINING AND SIMULATION

## Evaluation index

The selection of evaluation indicators should be combined with the needs of evaluation objects from a multi-level and multi-angle. For the construction of a talent evaluation index system in Colleges and universities, in addition to evaluating the knowledge reserve of evaluation objects, it is also necessary to assess the ability of evaluation objects in teaching and innovation ability to ensure the scientific and accurate evaluation results from multi-dimensional evaluation objects. This article constructs an evaluation index system for applied talent quality from four parts: knowledge cultivation level, innovation and practice ability, environmental adaptability and psychological and personality quality, as shown in Table 1.

**Table 1 Evaluation index system of applied talents.**

| Primary indicators | Secondary indicators |
|---|---|
| Knowledge level | Knowledge learning ability, education level, knowledge application ability and knowledge storage capacity |
| Innovation and practice ability | Imagination level, practical ability, attention, observation level, initiative in dealing with things, logical thinking ability, memory level |
| Environmental adaptability | Position ability, environmental pressure resistance ability, organizational cooperation ability, strain ability, their own behavior control ability |
| Psychology and personality quality | Social responsibility, dedication, professionalism, integrity, self-confidence, independent design ability, executive ability |

Note:
This article constructs an evaluation index system for the quality of applied talents from four parts: knowledge cultivation level, innovation and practice ability, environmental adaptability and psychological and personality quality.

## Data processing

In this article, the quantized data of each index is taken as the input matrix of DBN network, and all the input data are normalized. In this article, the Max-Min normalization method is used, that is, the indicator data is normalized to between [0,1], the maximum value is 1, and the minimum value is 0. The activation function in the training process uses the Sigmoid function. The specific calculation method is shown in Formula (10).

$$Y_i = \frac{x_i - x_{\min}}{x_{\max} - x_{\min}} \tag{10}$$

where $Y_i$ is the normalized data, $x_i$ is the actual value of the input vector. $x_{\max}$ and $x_{\min}$ are the maximum and minimum values in the input vector, respectively.

In order to more intuitively reflect the results of talent evaluation, the results are divided into four grades: excellent, medium and poor, and the results are expressed by 4-digit one hot code, with excellent being 1000, good being 0100, medium being 0010 and poor being 0001.

## Model training

### The number of hidden layers

DBN is a multi-layer deep network. The number of HLs directly affects the extraction ability of input features. In theory, the more layers of HLs are, the more complex the network structure is, the stronger the feature extraction's power and accuracy. However, with increased HLs, the training difficulty will gradually increase and the convergence speed will slow down. The MSE of output prediction data is taken as the evaluation standard. The experimental results are shown in Table 2, the average of 10 experiments.

The experimental results show that when the number of HLs is 2, the MSE of the test results is the smallest. Then, the number of nodes in each HL is determined by the ergodic method. Specifically, the number of nodes in the initial and second layers is established as [14, 15, 16, 17, 18, 19, 20] and [8, 10, 12, 14] respectively, resulting in a combination of 28 distinctive network structures. Each of these network structures is individually trained, and the comparative analysis of accuracy for every network model combination is depicted in Fig. 3.

| Table 2 Comparison of different HLs. | |
| --- | --- |
| **Number of HLs** | **MSE** |
| 1 | 0.31 |
| 2 | 0.22 |
| 3 | 0.26 |
| 4 | 0.28 |

**Note:**
DBN is a multi-layer deep network. The number of HLs directly affects the extraction ability of input features. In theory, it is considered that the more layers of HLs are, the more complex the network structure is, the stronger the ability and accuracy of feature extraction will be. However, with the increase of the number of HLs, the training difficulty will gradually increase and the convergence speed will slow down. The MSE of output prediction data is taken as the evaluation standard.

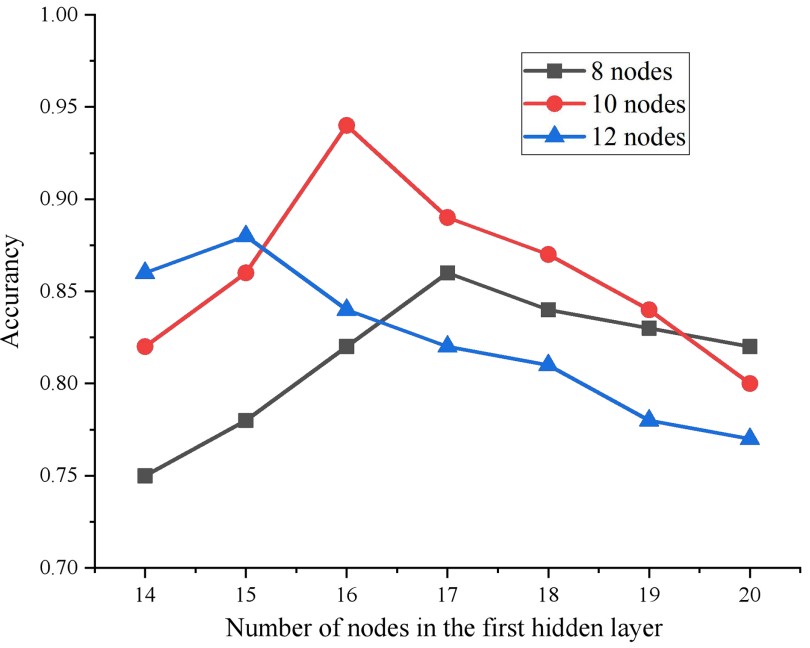

**Figure 3 Accuracy comparison of HL nodes with different combinations.**

By comparing different combinations, it is found that when the number of HL nodes of DBN is 16-10, The model achieves the best training effect. Therefore, the final network structure of this article is 28-16-10-4.

### Training error

The experiment mainly verifies the HS algorithm's and GAAHS algorithm's optimization effect on DBN. Therefore, both algorithms are optimized in the same DBN network structure. The variation of classification error rate with iteration times of the three models in fine-tuning training stage is shown in Fig. 4.

As seen in Fig. 4, with the increase in iteration times, the classification errors of the three models show a downward trend. After the number of backward propagation iterations of GAAHS-DBN is 20 times, the network begins to stabilize, while the HS-DBN and DBN
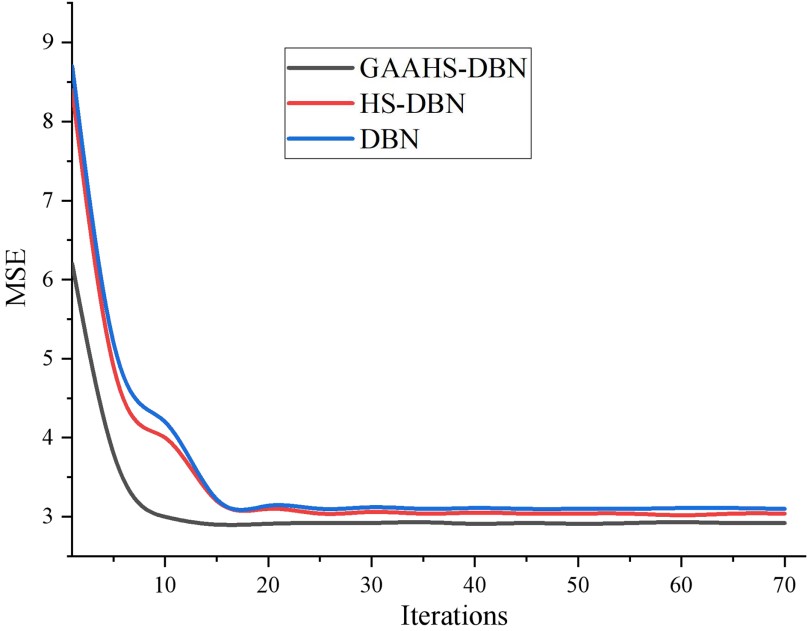

**Figure 4** **Error comparison in model training.**

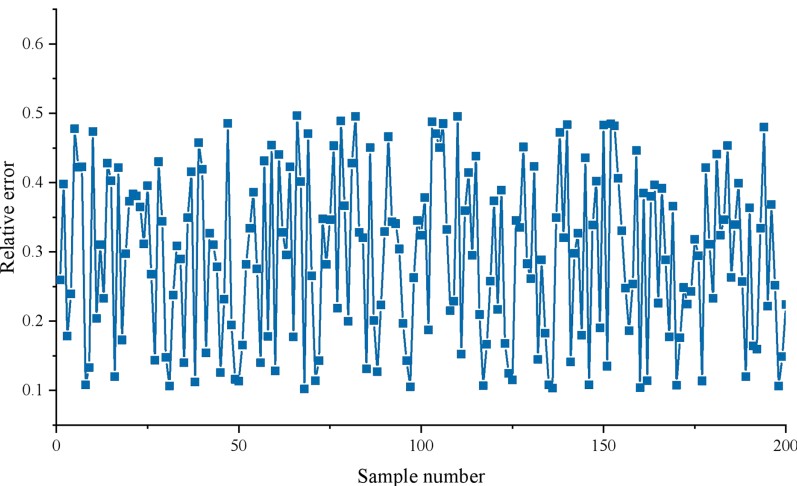

**Figure 5** **Prediction error results.**

models tend to be stable after 30 iterations. Therefore, under the same network structure, the convergence speed of the proposed model is the fastest, and the error rate of the final model is the smallest.

The test results are compared with the test data of the GAAHS-DBN model. This article's final evaluation result grade corresponds to the four output nodes and is represented by one hot code. Therefore, the actual evaluation level of each evaluation object is compared with the output value of the corresponding network node in the test process. Figure 5 shows the relative error between the model output and actual values.

As depicted in Fig. 5, the majority of the prediction relative errors of the model exhibit absolute values within the range of 0.2 to 0.4, indicating a high level of accuracy. Notably, this model employs the GAAHS algorithm to initialize the weights of the DBN. This approach effectively addresses the issue of DBN models often getting trapped in local optima due to random weight initialization. Furthermore, the utilization of GAAHS aids in fully leveraging the data information of applied talents, thereby enhancing the classification guidance of applied skills in practical teaching.

## CONCLUSION

For the transformation of talent training paths under the background of new liberal arts, this article puts forward a comprehensive evaluation model of applied talents based on improved DBN. This model uses the GAAHS algorithm to initialize the weights of DBN, which not only solves the problem that the DBN model is easy to fall into local optimum due to random initialization weights. The absolute value of the relative error for the model predominantly falls within the range of 0.1 to 0.4, indicating a high level of accuracy for the model. In addition, the global adaptive harmony adjustment method improves the local search ability of the algorithm, makes the results closer to the real optimal solution, and improves the model's performance. The training error of the model is small, which is helpful to fully excavate the data information of applied talents to enhance the classification guidance of applied talents in practical teaching. Higher model classification can broaden the way of talent demand information, create favourable conditions for professional construction, optimize the curriculum, innovate the assessment method and make other specific suggestions.

### Funding

The work is funded by 2022 Liaoning Province General Higher Education Undergraduate Teaching Reform Research Project "The construction and practice of the mode of "government, industry, university and research" collaborative cultivation of applied design talents under the background of new liberal arts construction". The funders had no role in study design, data collection and analysis, decision to publish, or preparation of the manuscript.

### Grant Disclosures

The following grant information was disclosed by the authors:
2022 Liaoning Province General Higher Education Undergraduate Teaching Reform Research Project.

### Competing Interests

The author declares that they have no competing interests.

## Author Contributions

- Fei Tang conceived and designed the experiments, performed the experiments, analyzed the data, performed the computation work, prepared figures and/or tables, authored or reviewed drafts of the article, and approved the final draft.

## Data Availability

The code is available in the Supplemental File.

The data is available at Kaggle:

https://www.kaggle.com/datasets/rhuebner/human-resources-data-set.

## Supplemental Information

Supplemental information for this article can be found online at http://dx.doi.org/10.7717/peerj-cs.1461#supplemental-information.

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
