# Peer review of "Construction of applied talents training system based on machine learning under the background of new liberal arts"

_PeerJ Computer Science, doi:10.7717/peerj-cs.1461_

## Round 0.1 · original submission · Minor Revisions

The paper must be modified based on the comment provided by the reviewers, so the paper can be published in PeerJ Computer Science

Reviewer 1 ·

Basic reporting

This paper proposes a comprehensive evaluation model of creative talents based on improved DBN. After the experiment, the training results showed that the optimized DBN on convergence speed and precision has a certain degree, on the creative talent evaluation classification accuracy is higher. However, there are also some problems, some revisions need to be revised. The common comments are as follows:

1. The author's logic in introducing the algorithm proposed in this paper is not coherent and complete. For example, in the introduction, after introducing the clustering algorithm, the author leads directly to the algorithm covered in this paper, which seems a bit abrupt. I suggest that the author add some background related to the algorithm as a transition.
2. The author has some problems with the literature review. For example, in Section 2.1, just listing the previous work makes it unclear to the readers what the author intends to do. It is suggested that the author should give a conclusion after presenting the previous work so that the reader can understand the current status and pain points of the existing research.
3. There are still some problems with the writing specifications of the figures. For example, the titles of Figure 1, Figure 2, Figure 3, Figure 4, and Figure 5 are not in the right place. The titles of the figures should be centered.
4. The author has some problems with the explanatory description of the figures. For example, the explanation of Figure 1 is not complete. The author does not explain all the contents of Figure 1, lacking both the “Student” and the “Enterprise Users”. I suggest that the author add these two parts.
5. There is a slight problem with the author's explanation of the formulas. Inadequate explanations make the readers more confused. I suggest that the author provide a detailed explanation of the variables of Formula 10.
6. Please make sure your 'conclusion' section underscore the scientific value added of your paper, and/or the applicability of your findings/results, as indicated previously.
7. Please revise your conclusion part into more details.

Experimental design

See above

Validity of the findings

See above

Additional comments

pl. see the above sections

Reviewer 2 ·

Basic reporting

The author has done a lot of work, including:
1 - In this model, the optimal result value generated by the iteration of GAAHS algorithm is used as the connection weight and bias of RBM, so that the weights and offsets have good initial values. in the pre-training stage of DBN.
2- The creative talented person's quality evaluation index system is constructed by four parts:
knowledge level, innovation practice ability, and psychological quality, the experiment yields satisfactory results. The manuscript can be accepted after minor modification.

Experimental design

The author should not simply list the previous work but should focus on its shortcomings to introduce the method proposed in this paper.
when introducing the methods, the author should also focus on explaining the necessity and importance of the research method to make the essay more logical.
In section 2.2, the author can briefly introduce the significance of the method before introducing its specific content
The author's contribution of figures and text needs to be improved. For example, I suggest that the author give a general introduction to the text before giving the overall design Figure 1 so that the readers can have an overall perception.
Please give more details about the theoretical foundations to support and further clarify these formulas.
The author should pay attention to the logical coherence of the context when writing. For example, in seciton 3.2.1, the author introduces the situation of " a smaller BW value is used",but gives a direct conclusion without introducing the situation of "the value of BW is larger"

Validity of the findings

The author needs to improve the standard of writing tables. For example, the titles of the 1 and Table 2 are not in the right place. The titles of the tables should be centered.
There is too much content in the conclusion. which looks disorganized in the menuscript. I sugges the authors re-organize such a section to make it more readable
There also same problem in language expression in this paper, which need to be modified. Pay attention to checking the puncuation in the text.

---

## Round 0.2 · accepted · Accept

The paper in very good standard, all comments have been addressed by the authors

Reviewer 1 ·

Basic reporting

The revised version of the paper is improved well, so I am satisfied with the current version of the paper and recommend it for further processing.

Experimental design

The experimental design in the manuscript seems to be of sufficient standard to be accepted.

Validity of the findings

I have no objections on the findings of the research and feel that they are okay

Additional comments

The revised version of the paper is improved well, so I am satisfied with the current version of the paper and recommend it for further processing.

Reviewer 2 ·

Basic reporting

The author has expressed all the previous comments with better improvements

Experimental design

Experimental design seems in a better condition now, it is acceptable

Validity of the findings

The overall improvements of author are better and I believe the paper should be accepted